# *Rickettsia amblyommatis* in Ticks: A Review of Distribution, Pathogenicity, and Diversity

**DOI:** 10.3390/microorganisms11020493

**Published:** 2023-02-16

**Authors:** Elise A. Richardson, R. Michael Roe, Charles S. Apperson, Loganathan Ponnusamy

**Affiliations:** Department of Entomology and Plant Pathology, North Carolina State University, Raleigh, NC 27695, USA

**Keywords:** *Rickettsia amblyommatis*, *Amblyomma* spp., *Amblyomma americanum*, lone star tick, tick-borne pathogens

## Abstract

*Rickettsia amblyommatis* is a potentially pathogenic species of *Rickettsia* within the spotted fever group vectored by ticks. While many studies have been published on this species, there is debate over its pathogenicity and the inhibitory role it plays in diagnosing illnesses caused by other spotted fever group *Rickettsia* species. Many publications have recorded the high infection prevalence of *R. amblyommatis* in tick populations at a global scale. While this species is rather ubiquitous, questions remain over the epidemiological importance of this possible human pathogen. With tick-borne diseases on the rise, understanding the exact role that *R. amblyommatis* plays as a pathogen and inhibitor of infection relative to other tick-borne pathogens will help public health efforts. The goal of this review was to compile the known literature on *R. amblyommatis*, review what we know about its geographic distribution, tick vectors, and pathogenicity, assess relatedness between various international strains from ticks by phylogenetic analysis and draw conclusions regarding future research needed.

## 1. Medical and Veterinary Importance of Ticks

Ticks and the pathogens they vector represent a significant threat to humans, livestock, companion animals, and wildlife. Ticks are responsible for transmitting a greater variety of pathogens than any other arthropod group [1,2]. Cases of tick-borne disease have been increasing within the United States, having doubled from >22,000 cases in 2004 to >48,000 cases in 2016, with Lyme disease accounting for 82% of these reports [3]. According to the CDC, reported tick-borne disease cases increased to 50,865 in the USA in 2019 [4]. Some of the tick-borne diseases that impact people include Lyme disease (*Borrelia burgdorferi*), Rocky Mountain spotted fever (*Rickettsia rickettsii*), tick-borne encephalitis (*Flavivirus*), babesiosis (*Babesia microti*), anaplasmosis (*Anaplasma phagocytophilum*), ehrlichiosis (*Ehrlichia ewingii* and *Ehrlichia chaffeensis*), and others [5,6]. Some of the most important species of ticks that are responsible for the spread of important human pathogens include *Amblyomma americanum*, *Amblyomma maculatum*, *Dermacentor variabilis*, *Ixodes pacificus*, *Ixodes ricinus*, *Ixodes scapularis*, *Rhipicephalus sanguineus*, and others [5,6,7]. These tick-borne diseases can be life-threatening, and while some vaccines have ongoing clinical trials, vaccine availability is currently lacking [8,9,10]. Tick-borne diseases are responsible for significant economic losses within the livestock industry and present a threat to public health worldwide [2]. The incidence of cases of tick-borne diseases is projected to increase as climate change continues to alter environmental conditions that may change the distribution of tick species [1,11,12,13]. This may be especially true for *rickettsial* spotted fever illnesses with ticks as their primary vector. Due to the enhancement of cell culture systems and molecular techniques, the number of identified species of *Rickettsia* has increased significantly over the past 40 years [14,15,16,17,18].

## 2. Materials and Methods

To write this review, five electronic databases (Web of Science, Google Scholar, PubMed, ScienceDirect, and SciELO) were searched during the time frame of 1 May to 5 December 2022 for relevant publications spanning multiple decades. To the best of our knowledge, we have written a review that is representative of all of the databases investigated. For this search, the following keywords were used: “*Rickettsia amblyommatis*”, “*Rickettsia amblyommii*”, and “*Candidatus* Rickettsia amblyommii”. The molecular phylogenetic analyses were completed using MEGA11 software (MEGA11: Molecular Evolutionary Genetics Analysis version 11) [19] using a maximum likelihood method on the Kimura 2-model (ML; bootstrap replicates: 1000). These blast searches were conducted on 3 February 2023.

## 3. Background on *Rickettsia*

The genus *Rickettsia* includes a diverse group of intracellular Gram-negative bacteria that are often the causative agents of disease for a variety of pathogens vectored by arthropods. Species within the genus of *Rickettsia* are split into four groups, including the spotted fever group (SFG), typhus group, *Rickettsia bellii* group, and the *Rickettsia canadensis* group [20,21,22]. It has also been recently shown that *Candidatus* Rickettsia mendelii is a separate basal group [23]. *Rickettsia* species are transmitted by a variety of arthropod vectors such as ticks, chiggers, fleas, and lice [24,25,26,27]. The maintenance of *Rickettsia* is dependent on animal hosts, mainly vertebrates; however, the role of these hosts in the life cycle of these bacteria is poorly understood [28]. Some hosts provide the necessary bloodmeal to ticks, and others provide essential supplemental components for the transmission of pathogens [28]. The spotted fever group (SFG) comprises the majority of tick-borne *Rickettsia*, consisting of 29 validly published species [29]. The primary reservoirs and vectors of SFG *rickettsia* are ticks. SFG *rickettsia* can move transstadially and transovarially, with some species of *Rickettsia* living within the entire life cycle of the tick [27,30]. Some species of *Rickettsia* are pathogenic and lethal causing serious morbidity in humans, while others are endosymbionts or symbionts not known to play an active role in human disease [30,31,32]. Some tick-borne *Rickettsia* species and the diseases they cause include *R. rickettsii*, *R. parkeri*, *R. akari* (spotted fever rickettsioses), *R. africae* (African tick bite fever), and *R. australis* (Australian tick typhus). In the United States, Rocky Mountain spotted fever (RMSF) caused by *R. rickettsii* is currently considered the most lethal identified rickettsial disease to humans [6,33,34]. Until 2010, most patients with tick-associated rickettsiosis were considered to have RMSF caused by *Rickettsia rickettsii* [35,36]. However, SFG rickettsial species elicit cross-reactive antibodies, and the immunofluorescence tests used by US public health labs cannot distinguish between the *Rickettsia* species that cause these different diseases [37]. Therefore, the CDC, as of 2010, now classifies cases formerly reported as RMSF as cases of SFG rickettsioses. 

While rickettsioses are some of the oldest known vector-borne diseases, there is still much that is unknown and even debated, especially regarding each species’ potential pathogenicity and status as being infectious or symbiotic [14,27]. With the arrival of molecular techniques, the understanding of *Rickettsia* has expanded. Still, there remain gaps in our knowledge, and with endemic rickettsioses cases on the rise, it is essential to understand these diseases fully. Over the last two decades, there has been great debate over the pathogenicity and significance of a specific ubiquitous species of *Rickettsia*, *Rickettsia amblyommatis*, which will be the focus of this review. 

### Background on Rickettsia amblyommatis 

*Rickettsia amblyommatis* (formerly *Candidatus* Rickettsia amblyommii) was first discovered in Tennessee, USA in 1973 from the tick, *Amblyomma americanum* [38,39]. After this initial finding, *R. amblyommatis* was discovered infecting *A. americanum* populations at high percentages in South Carolina, USA, and Arkansas, USA [38]. Since then, *R. amblyommatis* was discovered infecting tick populations at high prevalence worldwide, reaching greater than 90% in the USA and Panama [40,41,42]. While *R. amblyommatis* has primarily been isolated from ticks, it was recently discovered in chiggers (*Leptotrombidium peromysci* and *Eutrombicula* spp.) in North Carolina, USA [43]; however, this review will focus solely on ticks*. Rickettsia amblyommatis* was previously and informally described as *Rickettsia amblyommii* in the Stothard 1995 dissertation [44]. This species’ name as well as *Candidatus* R. amblyommii was used throughout publications until 2016 when Karpathy et al. [38] formally assigned the name, *R. amblyommatis*, to conform to the rules of the International Code of Nomenclature. 

Since its initial discovery, *R. amblyommatis* has proven to be an increasingly prevalent tick-borne rickettsial species, having been detected in 34 species of ticks in 17 countries (Figure 1; Table 1). Most significantly, *R. amblyommatis* has continually been detected at high prevalence in the USA in *A. americanum* populations, a geographically widespread human-biting tick [9,40,45,46,47,48]. In addition to being ubiquitous, *R. amblyommatis* is reported to have interesting other properties such as preventing ticks from acquiring other coinfected species of *Rickettsia* [31,49], altering tick host-seeking behavior [50,51], and affecting the progression of disease symptoms when infecting humans [52,53]. This is the first review of *R. amblyommatis*. In this review, we compiled all known information on *R. amblyommatis* in published studies, specifically discuss its distribution, discovery in different tick species, and pathogenicity, examined phylogenetic relatedness of the various strains of *R. amblyommatis* detected in ticks by PCR and from isolates of *R. amblyommatis* globally, and consider possible future directions of research needed.

## 4. Distribution and Spread of *Rickettsia amblyommatis*

### 4.1. Tick Vectors 

*Rickettsia amblyommatis* was reported in 34 species of ticks globally, with the majority within the genus *Amblyomma*. Figure 2 illustrates the proportion of each species of tick infected with *R. amblyommatis* reported in the literature. *Amblyomma* spp. made up about 83% of the ticks reported to be infected with *R. amblyommatis* (Figure 2). Through this literature search, *A. americanum* was found to be the most common species of tick infected with *R. amblyommatis*, making up about 27.8% of the reported tick species; *A. americanum* populations infection rates ranged from 1 to 90% (Table 1) [40,47,48,50,51,58,59,60,63,65,66,69,70,71,75,81,82,83,85,92,93,164]. Studies of the microbiome of ticks have found *R. amblyommatis* to be present in *A. americanum* and *A. maculatum*, specifically finding *R. amblyommatis* to be the most common member of Rickettsiales present within the lone star tick [164,165]. 

*Amblyomma americanum* is a human-biting tick and competent vector of the causative agents of RMSF, Human Monocytic Ehrlichiosis, tularemia, and heartland virus, and it is more recently known to cause a meat allergy in humans [45,166,167]. While *A. americanum* is native to the Southeastern United States, this tick is found to be significantly expanding in geographic range, perhaps due to a combination of its generalist tendencies and changing climatic conditions, which will likely further escalate the rise in tick-borne diseases [13,46,168,169,170,171]. This northward expansion has been so successful that *A. americanum* is now the most abundant species of tick in Long Island, New York [46,172,173,174]. This expansion and growth in populations could lead to shifts in disease prevalence by the possible displacement of other species of tick [80]. Due to *A. americanum*’s aggressive nature, successful expansion, and high rate of *R. amblyommatis* infection, it has been predicted that the number of infections of *R. amblyommatis* in people would increase with time [1,9,45,46,47,168]. However, in clinical settings, there is a lack of *Rickettsia* species-specific serological testing making it unclear if this prediction is occurring [34]. *Amblyomma longirostre* was the 2nd most common species of tick found to be infected with *R. amblyommatis*, making up about 10% of the infected species of tick reported. This species of tick has a geographic range from southern Mexico to Argentina with a wide distribution throughout Brazil [175]. 

Past studies have shown that ticks that are infected with a pathogen can experience changes in behavior, physiology, and survivability [176,177,178,179]. When a tick is infected with *R. amblyommatis*, the bacteria can be found in the ovaries, midgut, and salivary glands [9,180,181,182,183]. Ticks infected with *R. amblyommatis* displayed changes in their host-seeking behavior, such as spending less time questing compared to uninfected ticks [50,51]. It is unclear how this will affect the transmission dynamics and spread of *R. amblyommatis*. It is possible that infection with *R. amblyommatis* could cause a disruption in water balance dynamics in *A. americanum*, making ticks more susceptible to desiccation and preventing them from spending as much time host-seeking [51,177,184,185]. 

### 4.2. Geographic Range 

In the USA, *R. amblyommatis* has a wide geographic range, being identified in ticks as far south as Florida, as far west as Kansas, and as far north as New York [47,65] (Figure 1; Table 1). Outside of the United States, *R. amblyommatis* was detected in Brazil, Argentina, Australia, Belize, Chile, Panama, Colombia, Costa Rica, Cuba, Mexico, Nicaragua, Honduras, Paraguay, Pakistan, French Guiana, and El Salvador [97,102,103,110,111,118,126,141,142,147,149,150,153,156,159,160,163,186] (Figure 1; Table 1). In Table 1, the geographic range of *R. amblyommatis* for different tick species and the percentage infected in the population sampled are summarized. Additionally, Bermudez and Troyo (2018) concluded that this species was the most common rickettsial species in Central America [187]. 

In North Carolina, USA, it was concluded that *R. amblyommatis* is the most common rickettsial exposure that humans experience after being bitten by a lone star tick [40]. These data give validity to the suggestion that “*R. amblyommatis* likely represents the most prevalent and widely distributed SFG rickettsial species in the Americas” [38]. There are several plausible explanations for the widespread distribution of *R. amblyommatis*; however, one possible factor explaining the geographic expansion of *R. amblyommatis* is likely the resurgence and expansion of white-tailed deer populations; this mammal is the preferred host of *A. americanum* [45,48,188,189]. Since *A. americanum* is the most frequent carrier of *R. amblyommatis*, movement by deer may have led to the geographic expansion of *R. amblyommatis*. Ticks are considered both the reservoir and the vector for *R. amblyommatis* which causes quicker and further spread of this spotted fever species as it does not have to depend on hosts to proliferate [27,55,131,189,190] and, indeed, the geography of tick-borne rickettsioses such as *R. amblyommatis*, may be mainly based on the distribution of the vector tick species [131,190]. Another aspect that could be helping fuel this spread could be *Amblyomma* spp.’s generalist tendencies when finding a host, increasing their likelihood of survival by finding a host quickly and moving to the next life stage [191]. 

### 4.3. Hosts of Rickettsia amblyommatis-Infected Ticks

Workers in the majority of field studies that have detected *R. amblyommatis* in tick populations have collected unfed host-seeking ticks while dragging or flagging. However, in some cases, ticks were collected from wild or domestic animals. These data provide some information about possible vertebrate reservoir hosts of *R. amblyommatis* and could help determine the geographic distribution of this *Rickettsia* species. While some tick-borne bacterial pathogens have known reservoirs, *R. amblyommatis* does not; as stated earlier, some have hypothesized that there is no vertebrate reservoir [55,189]. For example, Apperson et al. (2008) reported white-tailed deer to be negative for antibodies to *R. amplyommii*, suggesting deer are not becoming infected with this rickettsial species by lone star ticks [189]. We have compiled a list of all of the animals found with feeding ticks infected with *R. amblyommatis* (Table 1). This list includes a wide variety of animals, with the most common including humans, wild birds, dogs, horses, deer, and others (Table 1). Birds in particular play an important role in tick-borne diseases by spreading pathogenic bacteria over long distances to new areas [122,147,192,193]. Studies on infected ticks feeding on birds have the potential to explain some of the distribution patterns of *R. amblyommatis*. In a study carried out by Dolz et al. [147], most *R. amblyommatis*-infected ticks were found on resident passerine birds that did not migrate. However, one *R. amblyommatis*-infected tick was found on *Catharus ustulatus*, a thrush that migrates far distances such as from the USA to Argentina. This finding reveals that migrating birds could potentially play a role in the dispersion of *R. amblyommatis* over long distances. Additionally, Budachetri et al. [194] investigated the microbiome of birds and the ticks infesting them and did not find similarities between the microbiomes, leading them to conclude that migratory birds do not act as a reservoir host for *R. amblyommatis* but rather could possibly play a role as a means of dispersal for ticks and tick-borne pathogens. Mukherjee et al. [193], who conducted a study on exotic ticks infesting migrating songbirds, found these ticks to harbor high levels of *Rickettsia* spp. including *R. amblyommatis* and concluded that it is likely that birds can transport these ticks and tick-borne pathogens long distances. 

## 5. Pathogenicity

There has long been conflicting information on whether *R. amblyommatis* is pathogenic to mammals. In 1981, Burgdorfer was unable to create rickettsial disease in guinea pigs when experimentally infected with *R. amblyommatis*; however, he found that this infection caused mild symptoms in meadow voles (*Microtus pennsylvanicus*) [55]. Despite that, the author concluded that *R. amblyommatis* was not pathogenic to humans due to the lack of symptoms such as a rash, headache, or fever reported by people in areas where *R. amblyommatis* tick infection prevalence was high. Since then, more studies have been conducted that have indicated that *R. amblyommatis* could induce illness in mammals. For example, mice that were intravenously infected with *R. amblyommatis* were found to have manifested mild disease, showing clinical symptoms such as weight loss [9]. However, it was seen that *R. amblyommatis* displayed an attachment deficiency to microvascular endothelial cells resulting in a slow growth rate, suggesting that mice would require high titers of *R. amblyommatis* to become infected and manifest symptoms of disease [9]. Similarly, Snellgrove et al. [195] found that when guinea pigs were inoculated with *R. amblyommatis*, some of the guinea pigs developed fever, orchitis, and dermatitis. Rivas et al. [196] found that when guinea pigs were inoculated with *R. amblyommatis*, the guinea pigs experienced an immune response in the form of antibody development, and two out of six of the animals displayed testicular alterations; however, no fever, weight loss, or other signs of infection were exhibited. Levin et al. [180] reported that guinea pigs developed a disseminated infection when fed upon by ticks infected with *R. amblyommatis*. *Rickettsia rickettsii* has also been shown to have highly varied levels of virulence in animal models; therefore, it is not surprising that this is the case with *R. amblyommatis* [9,197,198]. It is also possible that these differences in observed pathogenicity in guinea pigs could be due to varied strains of *R. amblyommatis* being used in the experiments [9]. 

While most researchers did not conduct serological testing on animals with attached ticks, there were some exceptions as well as studies on animals artificially infected with *R. amblyommatis* to evaluate transmission potential. Some studies did not report any clinical symptoms but were able to demonstrate the transmission of *R. amblyommatis* from infected ticks to the animals by detecting seroactivity, such as antibody titers to *R. amblyommatis* in rural dogs (*Canis familiaris*), rabbits (*Oryctolagus cuniculus*), white-eared possums (*Didelphis albiventris*), the black rat (*Rattus rattus*), and *R. amblyommatis* DNA was found in a single case in the blood of a dog [84,114,131,133,134,146,186,199]. The majority of the results from these mammal studies show a consistent pattern that supports the premise that *R. amblyommatis* may cause a self-limiting mild febrile illness in mammals and humans, with immune-compromised individuals being particularly vulnerable to infection with this pathogen [9,196]. 

### 5.1. Rickettsia amblyommatis in Humans

Little information is available on cases of *R. amblyommatis* infection in humans. While there are increasing numbers of cases of RMSF-like illnesses, there is a lack of detection of *R. rickettsii* in tick surveillance studies, suggesting that these illnesses are associated with the expanding range of *A. americanum* and possibly the transmission of *R. amblyommatis* [46,80]. *Rickettsia amblyommatis* being a prevalent bacterium does not necessarily prove that this species is causing disease in humans; however, studies mentioned previously have implicated that infection with *R. amblyommatis* is correlated with symptoms of disease. For example, one human patient was found to have a rash at a tick bite site, and PCR testing on the tick found it to be infected with *R. amblyommatis*, making this the first known association of *R. amblyommatis* with a rash [52]. The sequencing data obtained had an identical match to *R. amblyommatis*, accession number EF063690 [52]. Billeter et al. [52] discussed a situation where soldiers became ill with febrile symptoms after working in *A. americanum*-infested sites and stated that serological evidence indicated that infection with *R. amblyommatis* was the possible cause. In Georgia, USA, twelve residents experiencing tick bites met the criteria for a possible tick-borne illness by having ticks attached for 6 h or more and manifested clinical symptoms of a tick-borne illness after a 4-d incubation [53]. These residents obtained PCR testing of their attached ticks and eight were found to be infected with *R. amblyommatis*. Two of these individuals who had an *R. amblyommatis*-infected tick bite reported a bulls-eye rash (erythema migrans) [53]. However, these individuals had multiple ticks attached to them, and it could not be determined which tick was originally at the site of the rash. In North Carolina, USA, outdoor workers are experiencing an increased risk of tick-borne infections; in one serologic study, seroconversions to *R. amblyommatis* were demonstrated, indicating infection, but no symptoms were recorded in the workers [200]. While there have not been any cases where *R. amblyommatis* was isolated from human patients, serological evidence indicates that this species of *Rickettsia* could be the cause of human illnesses displayed as RMSF-like symptoms [9]. 

Apperson et al. [189] collected ticks in North Carolina, USA, where human cases of RMSF were high. The collected ticks were molecularly tested for bacterial pathogens and 11 out of 25 *A. americanum* pools were found to be positive for *R. amblyommatis* [189]. Human patients from the same county that were presumptively diagnosed with a tick-borne illness were serologically tested, and three out of six patients were diagnosed as probable RMSF cases showing a greater increase in IgG-class antibody titers between paired acute and convalescent sera to *R. amblyommatis* antigens but not to *R. rickettsii* antigens [189]. These results suggest that human cases of spotted fever rickettsiosis diagnosed as RMSF may actually be experiencing an infection with *R. amblyommatis*. The authors felt that further investigations were needed into the pathogenicity of *R. amblyommatis* in humans. Other studies have also suggested that *R. amblyommatis* is a human pathogen causing misdiagnoses with other rickettsial spotted fever illnesses such as RMSF [47,201,202].

The studies that have been discussed show that it is possible that *R. amblyommatis* is pathogenic and can result in clinical symptoms in humans and animals. However, an infection with this pathogen may result in a minor, self-limiting illness in most individuals, especially those who are immune compromised [9,196]. It is also worth noting that when attempting to diagnose rickettsial infections, there may be confounding effects as all SFG rickettsia induce cross-reading antibodies [35]. With that in mind, previous infection with *R. amblyommatis* could be mistakenly diagnosed as RMSF during serological testing. Another hypothesis is that simultaneous infection with *R. amblyommatis* and other spotted fever group rickettsiae species would result in a lack of clinical symptoms due to this coinfection providing cross-immunity [145,147]. Other species of *Rickettsia* have overlapping geographic ranges with *R. amblyommatis*, such as *R. parkeri*, which is considered pathogenic to humans [203]. This could result in issues during diagnosis. While there is serological confirmation of ticks infected with *R. amblyommatis* that have bitten humans, there is still no direct evidence that *R. amblyommatis* may cause human infection.

### 5.2. Coinfection Effects and Inhibition

Over 25 years ago, Telford et al. [204] warned of the unique impact coinfections could have on public health with a growing number of highly widespread diverse tick-borne pathogens. With *A. americanum* experiencing a markedly expanding geographic range, this could affect the likelihood of coinfections. Recent studies have shown that when *R. rickettsii* (RMSF) is coinfected with *R. amblyommatis*, the latter may act as a sort of protection, lowering the severity of RMSF through an elicited immune response. Blanton et al. [31] found that guinea pigs who were initially inoculated with *R. amblyommatis* did not become ill when exposed to *R. rickettsii* but guinea pigs without previous exposure to *R. amblyommatis* did become ill from *R. rickettsii*. This study provided evidence for the potential protective role that *R. ambyommatis* plays in the epidemiology of RMSF. A similar finding was reported by Wright et al. [49], where *A. americanum* was infected with *R. amblyommii* and was less likely to develop an *R. parkeri* infection when feeding alongside an *R. parkeri*-infected *A. maculatum*. Additionally, it was seen that *R. amblyommatis*-infected *A. americanum* were less likely to become infected nor maintain an infection of *R. parkeri* [49]. These studies show that it is likely that *R. amblyommatis* is displaying a form of rickettsial exclusion, making coinfection with another rickettsial species less likely [55,139]. These findings suggest that an increase in infection prevalence of *R. amblyommatis* in ticks might decrease cases of RMSF. Interestingly, after two decades of increasing cases of RMSF, in 2018 and 2019 there was a slight decrease in reported cases of RMSF [34]. Similar protective effects when *R. amblyommatis* is coinfected with other *Rickettsia* species might also occur; *A. americanum*, the main transmitter of *R. amblyommatis*, also transmits other pathogens [205]. Investigations into the effects of *R. amblyommatis* coinfection with other tick-borne pathogens affecting virulence, transmissibility, and clinical symptoms in hosts are clearly warranted. 

## 6. Phylogenetic Analysis

The cultivation of *Rickettsia* species can vary depending on the cell line and culture system used. Sayler et al. [59] found that when culturing tick-borne organisms, it was best to use the cell lines acquired from the tick vector itself. *Rickettsia amblyommatis* has been propagated in the past in tick cell lines, mosquito cell lines, Vero cells, tissue cells, chicken fibroblast, primary embryonated chicken eggs, and the HUVEC line [38,59,69,135,206]. Karpathy et al. [38] established a pure culture of strain WB-8-2T (=CRIRC RAM004T=CSURP2882T) and assigned it as the type strain for *R. amblyommatis*. Prior to this, the species was isolated from *A. americanum* by Stothard et al. [44]. It was seen that this isolate was similar to other species of *Rickettsia* but proved to be unique through the analysis of the 17 kDa antigen gene [38]. The genome of *R. amblyommatis* of the strain GAT-3OV has been sequenced and submitted under the accession number NC_017028. The phylogenetic analysis completed by Karpathy et al. [38] was concatenated using five gene regions (*rrs*, *gltA*, *sca0*, *sca5*, and *sca4*) and showed that the most closely related *Rickettsia* species to *R. amblyommatis* were *R. japonica* and *R. raoultii*.

Past studies have hypothesized that various strains of *R. amblyommatis* could be genetically independent, with differences evolving based on the tick species and geographic location [99,190]. Dolz et al. [147] found that Costa Rican strains of *R. amblyommatis* showed intraspecific variation and found high associations with strains from some South American countries. With that in mind, we sought to create a phylogenetic tree of the *R. amblyommatis* sequences that were present in the NCBI database to compare strains and investigate the relationship between strains, location, and tick species. We created two trees, one that focused on the gene targeting the outer membrane protein A (*ompA*) and a second targeting outer membrane protein B (*ompB*). Sequences focusing on these two gene regions were selected as they have been noted to be polymorphic, with *ompB* being suitable for analyzing intraspecific differences in *R. amblyommatis* [190]. The methods involved in the creation of the phylogenetic trees can be seen in the Materials and Methods section. 

In total, there were 104 sequences identified for *ompA*; however, 25 (24.04%) were not included due to their small size; only sequences greater than 406 base pairs were included in the analysis. The pairwise comparison of *ompA* sequences revealed similarity values between 97.27% and 100% among strains. For the analysis of *ompB*, 19 sequences were incorporated, and only sequences greater than 600 base pairs were included for a total of 618 base pairs in the final tree. The pairwise comparison of *ompB* sequences revealed similarity values between 98.2% and 100% among strains. 

Phylogenetic analysis based on *ompA* gene sequences with maximum likelihood revealed four major branches (groups) among the *R. amblyommatis* strains (Figure 3). Group 1 consisted of 56 sequences, group 2 comprised ten strains of *R. amblyommatis*, group 3 contained ten sequences of *R. amblyommatis*, and group 4 had two sequences. The *ompA* phylogenetic analysis showed that *R. amblyommatis* did not display cluster organization for clusters 1,2, and 4 but did for cluster 3. Cluster 3 appears to be organized by tick species and geographic origin of the sequences where most of the sequences came from Brazil, with two being from Costa Rica and Colombia, and the majority of the sequences were obtained from *A. longirostre*. The lack of cluster organization in the other clusters could be due to the sequence length in this analysis (Figure 3). For example, in group 1 (N = 56), *R. amblyommatis* (Aa) sequences were identified to include as many as 14 different tick species from over eight different countries. Phylogenetic analysis of the *ompB* gene formed two groups (Figure 4). Group 1 contains 17 sequences and group 2 contains two sequences. These clusters do not appear to be organized by location or species of tick. 

## 7. Conclusions and Future Directions

While there is still much that is unknown about *R. amblyommatis*, through the compilation of published work on this topic, we can see that some peer-reviewed studies have found *R. amblyommatis* to be mildly pathogenic to mammals and humans, that it influences the virulence of other pathogens when coinfected, and that there is intraspecific variation between strains of this pathogen. However, overall, there are inconsistencies, with some studies not finding proof of pathogenicity and others finding symptoms of disease associated with *R. amblyommatis*. Disease symptoms from infection have been seen in humans and laboratory animals; however, some laboratory challenge studies could not produce similar findings*. Rickettsia amblyommatis*’s status as a pathogen needs to be verified through further testing. 

There remain many questions about *R. amblyommatis*, which could lead to productive research. One of the most important gaps in the knowledge is how coinfections with *R. amblyommatis* affect the virulence and the symptoms caused by coinfection with pathogens other than *R. rickettsii*. This could reveal if *R. amblyommatis* consistently diminishes the effects of other pathogens in coinfection or if this is only the case with *R. rickettsii*. It is also possible that a previous infection with *R. amblyommatis* could elicit antibodies that diminish the pathogenic effects of other SFG rickettsiae. Current common serological testing available in clinical settings lacks the resolution to detect the different SFG rickettsias, which prevents further understanding of the effects and distribution of *R. amblyommatis* in humans [34,201]. Studying the various strains of *R. amblyommatis* could elucidate if the genetic variability between the different strains leads to differences in the virulence or symptoms caused by each strain. Future studies are required that culture *R. amblyommatis* from different geographic regions and hosts as well as, elucidate the whole genome to characterize different isolates. This may lead to a better understanding of the diversity of *R. amblyommatis* strains, and perhaps their virulence. This research will facilitate the development of more specific pathogen-specific diagnostics. 

## Figures and Tables

**Figure 1 microorganisms-11-00493-f001:**
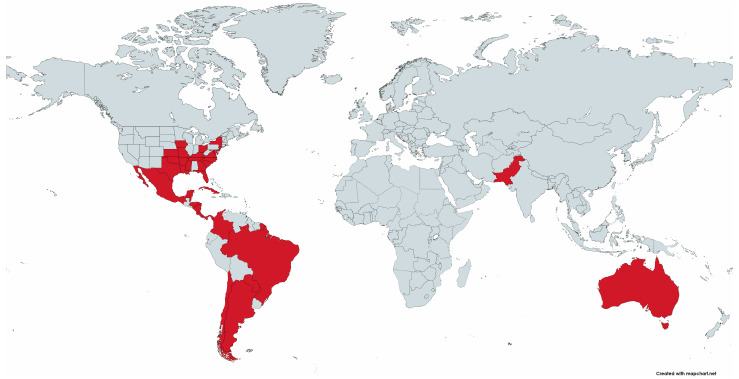
World map displaying the distribution of *Rickettsia amblyommatis*-infected ticks. Map created using www.mapchart.net (accessed on 27 January 2023).

**Figure 2 microorganisms-11-00493-f002:**
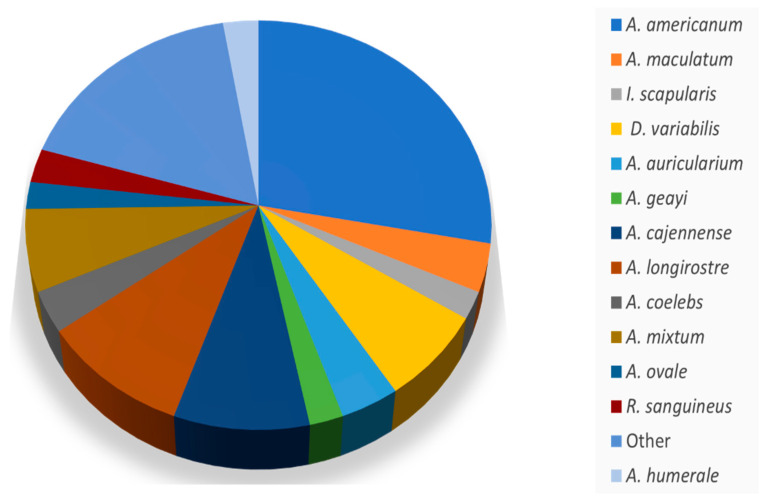
A pie graph of the species of tick reported infected with *Rickettsia amblyommatis.* The legend labeled as “other” included species that were only seen reported in 1–3 instances. These species were as follows: *A. breviscutatum*, *A. calcaratum*, *A. dubitatum*, *A. hadanii*, *A. humerale*, *A. inornatum*, *A. neumanni*, *A. oblongoguttatum*, *A. pacae*, *A. parvum*, *A. pseudoconcolor*, *A. rotundatum*, *A. tonelliae*, *A. varium*, *D. nitens*, *H. anatolicum*, *H. juxtakochi*, *Hyalomma dromedarii*, *Ornithodoros* sp., *I. tasmani*, and *R. microplus*.

**Figure 3 microorganisms-11-00493-f003:**
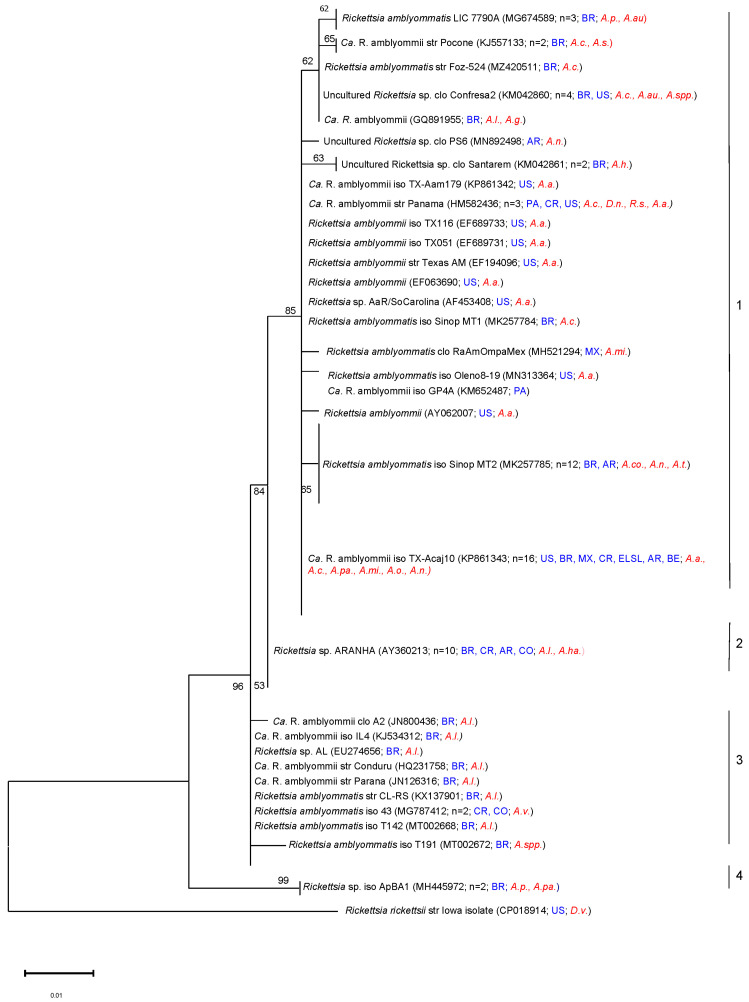
Phylogenetic tree of published sequences and reports of *Rickettsia amblyommatis* in ticks. The analyses were based on the sequences of the *ompA* gene region of *Rickettsia amblyommatis* within 406 bp obtained from NCBI BLAST. This tree shows a collapsed version incorporating a total of 79 nucleotide sequences (n shows the total number of sequences included in that group). The phylogenetic analysis was completed using a maximum likelihood method on the Kimura 2-parameter model (ML; bootstrap replicates: 1000). This tree is collapsed with nodes organized by sequence differences of 0.001. Instances in which publications reported the species as “*Rickettsia* spp.” but indicated in the publication that the species was *R. amblyommatis* were included in this analysis. The GenBank accession numbers are given in parentheses. Scale bars indicate the number of substitutions per nucleotide position. The *Rickettsia rickettsii* Iowa isolate (CP018914) was included as an outgroup. Tick species included in this tree are as follows: *A auricularium (A.au.)*, *A. cajennense (A.c.)*, *A. coelebs (A.co.)*, *A. geayi (A.g.)*, *A. hadanii (A.ha.)*, *A. humerale (A.h.)*, *A. longirostre (A.l.)*, *A. americanum (A.a.)*, *A. mixtum (A.mi.)*, *A. neumanni (A.n.)*, *A. oblongoguttatum (A.o.)*, *A. pacae (A.pa.)*, *A. parvum (A.par.)*, *A. pseudoconcolor (A.p.)*, *A. sculptum (A.s.)*, *Amblyomma* spp*. (A.spp.)*, *A. varium (A.v.)*, *D. nitens (D.n.)*, *D. variabilis (D.v.)*, and *R. sanguineus (R.s.)*. The countries included in this tree are as follows: Brazil (BR), United States of America (US), Mexico (MX), Panama (PA), Argentina (AR), Costa Rica (CR), Belize (BE), El Salvador (ELSL), and Colombia (CO).

**Figure 4 microorganisms-11-00493-f004:**
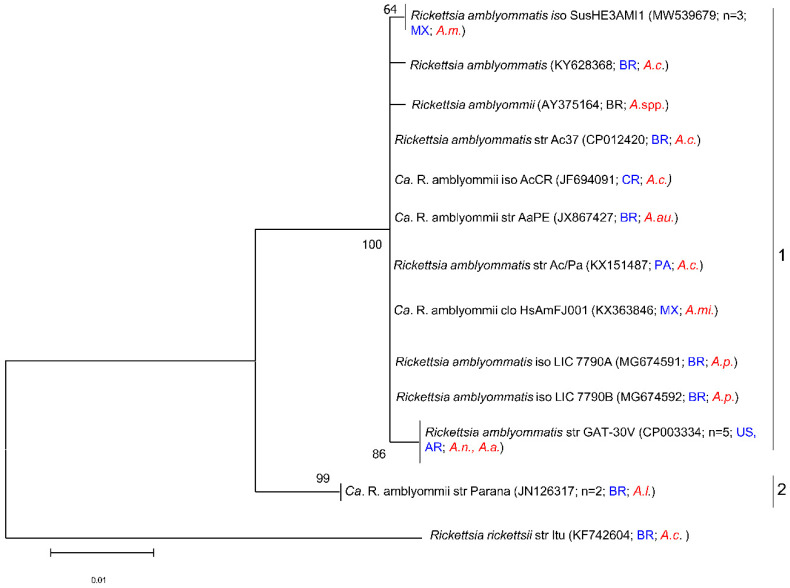
Phylogenetic tree of published sequences and reports of *Rickettsia amblyommatis* in ticks. The analyses were based on 20 partial sequences of the *ompB* gene region of *Rickettsia amblyommatis* within 618 bp obtained from NCBI BLAST. The phylogenetic analysis was completed using a maximum likelihood method on the Kimura 2-parameter model (ML; bootstrap replicates: 1000). This tree is collapsed with nodes organized by sequence differences of 0.001 (n shows the total number of sequences included in that group). Instances in which papers reported the species as “*Rickettsia* spp.” but indicated in the publication that the species was *R. amblyommatis* were included in this analysis. The GenBank accession numbers are given in parentheses. Scale bars indicate the number of substitutions per nucleotide position. This phylogenetic tree was created using MEGA11. Tick species included in this tree are as follows: *A. americanum (A.a.)*, *A auricularium (A.au.)*, *A. cajennense (A.c.)*, *A. longirostre (A.l.)*, *A. maculatum (A.m.)*, *A. mixtum (A.mi.)*, *A. neumanni (A.n.)*, *A. pseudoconcolor (A.p.)*, and *Amblyomma* spp*. (A.spp.).* The countries included in this tree are as follows: Brazil (BR), the United States of America (US), Panama (PA), Argentina (AR), Mexico (MX), and Costa Rica (CR).

**Table 1 microorganisms-11-00493-t001:** Summary of quantitative data obtained from peer-reviewed publications for *R. amblyommatis*-infected wild ticks collected globally. This table includes the location in which they were collected, the species of tick, the infection prevalence, and the collection type. Studies were included if *R. amblyommatis* was confirmed through conventional PCR and sequencing analysis.

Location	Species of Tick	Infection Prevalence % ^a^	Collection Type	Reference
AR, USA	*A. maculatum*	28% (58/207)	Off-host (white-tailed deer, dogs)	[54]
AR, USA	*A. americanum*	41.9% (463) ^b^	Vegetation	[55]
AR, USA	*A. americanum*, *A. maculatum*, *I. scapularis*, *Amblyomma* spp.	NA ^c^	Off-host (white-tailed deer, dogs)	[56]
AR, USA	*A. americanum*	98% (42/43 pools)	Flagging, CO_2_ Trapping	[57]
FL, USA	*A. americanum*	29% (391/1312)	Flagging	[58]
FL, USA	*A. americanum*	29.5% (52/176)	Dragging	[50]
FL, USA	*A. americanum*	38.0% (223/588)	Dragging	[51]
FL, USA	*A. americanum*	57.1% (845/1479)	Flagging	[59]
FL, USA	*A. americanum*	37.1% (56/151)	Flagging	[47]
FL, USA	*A. americanum*, *D. variabilis*	57.3% (63/110), 35% (6/17)	Off-host (wild pig)	[60]
FL, USA	*A. americanum*, *A. maculatum*	27.8% (5/18), 16.7% (1/6)	Black bears	[61]
GA, USA	*A. americanum*	0.52% (1/194)	Dogs	[62]
GA, USA	*A. americanum*, *D. variabilis*, *A. maculatum*	27.5% (117/426), 1.4% (2/142), 9.1% (4/22)	Off-host (humans)	[53]
GA, USA	*A. americanum*	87% of pools (519/598)	Flagging	[63]
GA, USA	*A. americanum*	44.7% (315/704)	Flagging	[47]
Guam, Marianna Islands, USA	*A. breviscutatum*, *R. microplus*	2.67% (3/112)	Off-host (wild pig, Philippine deer)	[64]
IA, USA	*A. americanum*	57.9% (11/19)	Flagging	[47]
KS, USA	*A. americanum*, *D. variabilis*	93% nymphal pools, 94% adult pools, and 4.7% (169)	Dragging	[65]
KY, USA	*A. americanum*, *D. variabilis*	27.8% (30/108), 1.1% (2/179)	Off-host (canine, hog, horse, raccoon, white-tailed deer, human)	[66]
KY, USA	*D. variabilis*	0.8% (1/124)	Unknown ^e^	[67]
LA, USA	*A. americanum*	60% (3/5)	Off-host (black bears)	[41]
MD, USA	*A. americanum*	87.9% (29/33 pools), 90% (27/30 individuals)	Flagging	[68]
MD, USA	*A. americanum*	64.50% ^d^	Dragging	[69]
MO, USA	*A. americanum*	3.2% (40/1269)	Field surveillance	[70]
MO, USA	*A. americanum*, *D. variabilis*, *I. scapularis*	89% (24/27 pools), 100% (7/7 pools), 50% (1/2)	Flagging	[71]
MO, USA	*A. americanum*	74% (213/288)	Dragging, CO_2_ bait traps, Off-host (human)	[72]
MS, USA	*A. americanum*	92.4% (73/79)	Dragging	[73]
NC, USA	*A. americanum*, *A. maculatum*, *D. variabilis*, *I. scapularis*	90.9% (412/453), 28.6% (2/7), 20% (1/5), 8.3% (1/12)	Off-host (humans)	[40]
NC, USA	*D. variabilis*	29.3% (156/532)	Flagging	[74]
NC, USA	*A. americanum*, *D. variabilis*	56.4% (871/1590), 11.1% (4/36)	Flagging	[75]
NC, USA	*A. americanum*	55.2% (216/391)	Flagging	[47]
Northeast, USA	*D. variabilis*	1.7% (3/181)	Off-host (dogs)	[76]
NJ, USA	*A. americanum*	6.6% (8/121)	Flagging	[47]
NJ, USA	*A. americanum*	25% (465/1858)	Dragging	[77]
NJ, USA	*A. americanum*	20% (49/245)	Sweeping vegetation	[78]
NJ, USA	*A. americanum*	12.8% (36/281)	Dragging and standard walking	[79]
NY, USA	*A. americanum*	58.35% (394/676)	Flagging	[80]
NY, USA	*D. variabilis*	8.3% (1/12)	Unknown ^e^	[67]
NY, USA	*A. americanum*	41.7% (198/475)	Flagging	[47]
OH, USA	*A. americanum*	30.2% (93/308)	Off-host (humans), Flagging	[81]
OH, USA	*A. americanum*	38% (8/21)	Humans	[82]
OK, USA	*A. americanum*	10% (6/60)	Flagging	[47]
OK, USA	*A. americanum*	33.6% (146/434)	Unknown ^e^	[83]
OK, USA	*A. americanum*	17.6% (123/700)	Dry Ice Traps	[84]
RI, USA	*A. americanum*	47.4% (18/38)	Flagging	[47]
SC, USA	*A. americanum*	11.7% (545) ^b^	Off-Vegetation	[55]
SC, USA	*A. americanum*	45.6% (36/79)	Flagging	[47]
SC, USA	*A. americanum*, *D. variabilis*	80% (84/105), 66.7% (10/15)	Off-host (Wild Pig)	[60]
TN, USA	*A. americanum*	16.6% (96) ^b^	Vegetation	[55]
TN, USA	*A. americanum*, *D. variabilis*	40% (255/655), 2.5% (14/555)	Off-host (humans, wild animals, dogs)	[85]
TN, USA	*A. americanum*	4.9% (45/926)	Dragging, CO_2_ baited trap	[86]
TX, USA	*A. auricularium*, *A. geayi*, *A. longirostre*	28.6% (2/7), 33.3% (1/3), 11.9% (5/42)	Off-host (wild birds)	[87]
TX, USA	*A. americanum*, *I. scapularis*	50% (41/82), 1.47% (2/136)	Dragging	[88]
TX, USA	*A. americanum*	100% (68/68 pools)	Dry ice trapping, Dragging, and Flagging	[89]
TX, USA	*A. inornatum*	100% (3/3)	Off-host (deer)	[90]
TX, USA	*A. inornatum*	Unknown ^g^	CO_2_ Traps	[91]
TX, USA	*A. americanum*, *A. cajennense*	30.3% (179), 32.3% (10) ^f^	Off-host (humans)	[92]
VA, USA	*A. americanum*	72.8% (pools)	Dragging	[93]
VA, USA	*A. americanum*	80.3% of pools ^d^	Flagging	[94]
VA, USA	*D. variabilis*	4.2% (9/214)	Unknown ^e^	[67]
VA, USA	*A. americanum*	19% (65/340)	Dragging	[95]
USA	*D. variabilis*	0.1% (7/5286)	Off-host (humans)	[96]
USA	*A. americanum*	80.5% (58/72 pools), 66.5% (244/367)	Off-host (humans)	[48]
Córdoba, Argentina	*A. neumanni*	23.6 (13/55)	Walking Survey, CO_2_ traps	[97]
Northern Argentina	*A. neumanni*	21% (3/14)	Off-host and Flagging	[98]
Salta, Argentina	*A. hadanii*, *A. neumannii*	21.3% (27/127), 44.4% (8/18)	Dragging	[99]
Salta, Argentina	*A. tonelliae*	1.47% (1/68)	Dragging	[100]
Argentina	*A. hadanii*	1.6% (1/60)	Vegetation and Off-host (cattle)	[101]
Australia	*I. tasmani*	28.2% (22/78)	Off-host (koalas)	[102]
Belize	*A. pacae*	100% (1/1)	Off-host (spotted paca)	[103]
Belize	*A. cajennense*, *A. maculatum*	55.1% (81/147), 66.7% (2/3)	Off-host (dog)	[104]
Acre, Brazil	*A. humerale*	33.3% (1/3) of A. humerale	Off-host (capybara)	[105]
Acre, Brazil	*A. longirostre*, *A. geayi*	62.5% (5/8), 50% (1/2)	Off-host (wild birds)	[106]
Acre, Brazil	*A. rotundatum*	14.3% (1/7)	Off-host (Amazon tree boa)	[107]
Amazonas, Brazil	*A. humerale*	2.17% (1/46)	Off-host (tegu)	[108]
Bahia, Brazil	*A. longirostre*	15.4% (2/13)	Off-host (wild birds)	[109]
Bahia, Brazil	*A. longirostre*, *A. varium*	24.4% (11/45), 20% (1/5)	Off-host (wild birds)	[110]
Bahia, Brazil	*A. longirostre*	23.7% (9/38)	Off-host (thin-spined porcupine and hairy dwarf porcupine)	[111]
Bahia, Brazil	*A. auricularium*, *Ambylomma* spp. larvae	57.5% (46/80)	Off-host (gray short-tailed opossum, rodent, Brazilian three-banded armadillo)	[112]
Espírito Santo, Brazil	*A. humerale*	50% (5/10)	Off-host (tortoise)	[113]
Maranhão, Brazil	*A. cajennense*	1% (1/100)	Off-host (dogs)	[114]
Maranhão, Brazil	*A. cajennense*	16.7% (3/18)	Off-host (horses)	[115]
Mato Grosso, Brazil	*A. cajennense*	0.45% (3/665)	Off-host (horse, donkey)	[116]
Mato Grosso, Brazil	*A. cajennense*	31% (5/16 pools)	Off-host (human) and flagging	[117]
Mato Grosso, Brazil	*A. auricularium*, *A. cajennense*	100% (1/1), 69.4% (136/196)	Off-host (wild animals)	[118]
Mato Grosso, Brazil	*A. cajennense*, *A. ovale*	3.3% (5/152), 15.4% (2/13)	Off-host (dogs)	[119]
Mato Grosso, Brazil	*A. cajennense*, *A. coelebs*	38.4% (10/26), 70% (7/10)	Off-host	[120]
Mato Grosso, Brazil	*A. sculptum*	1.2% (1/85 pools)	Off-host (Azara’s agouti)	[121]
Mato Grosso, Brazil	*A. longirostre*	45.5% (5/9)	Off-host (wild birds)	[122]
Mato Grosso, Brazil	*A. dubitatum*	5% (1/20)	Off-host (dog)	[123]
Mato Grosso, Brazil	*A. cajennense*, *A. coelebs*, *A. humerale*, *Amblyomma* spp.	NA ^g^	Off-host (rodents, marsupials)	[124]
Minas Gerais, Brazil	*A. longirostre*	(13/49; 26%)	Off-host (wild birds)	[125]
Pará, Brazil	*A. longirostre*, *A. humerale.*	70% (7/10), 22.2 % (2/9)	Off-host (wild animals)	[118]
Pará, Brazil	*A. longirostre*, *A. geayi*	56.7% (38/67), 57.1% (4/7)	Off-host (birds)	[126]
Paraíba, Brazil	*A. auricularium*, *A. longirostre*	23.1% (3/13), 45.5% (10/22)	Off-host (wild birds)	[127]
Paraná, Brazil	*A. longirostre*	32.3% (11/34)	Off-host (wild Birds)	[128]
Paraná, Brazil	*A. coelebs*	1.9% (8/420)	Off-host (ring-tailed coatis)	[129]
Pernambuco, Brazil	*A. auricularium*	20% (1/5)	Off-host (rodent)	[130]
Pernambuco, Brazil	*A. auricularium*	53.3% (8/15)	Off-host (skunk, rabbit)	[131]
Pernambuco, Brazil	*A. pseudoconcolor*	90.9% (10/11)	Off-host (dog, six-banded armadillo)	[132]
Rio Grande do Norte, Brazil	*A. auricularium*	11.1% (14/126 individuals and pools)	Off-host (white-eared opossum and xix-banded armadillo)	[133]
Rio Grande do Sul, Brazil	*A. longirostre*	66.7% (2/3)	Off-vegetation, Dragging	[134]
Rondônia, Brazil	*A. coelebs*, *A. cajennense*	14.3% (1/7), 26.8% (11/41)	Off-vegetation	[135]
Rondônia, Brazil	*A. cajennense*, *A. oblongoguttatum*	6.3% (2/32) + 2 pools, 16.7% (1/6)	CO_2_ traps; drag flagging; off-vegetation	[136]
Santa Catarina, Brazil	*A. longirostre*	100% (1/1)	Unknown ^e^	[137]
São Paulo, Brazil	*A. coelebs*	Unknown ^g^	Off-vegetation	[138]
Brazil	*A. calcaratum*, *A. longirostre*	50% (1/2), 12.3% (9/73)	Off-host (wild Birds)	[139]
Brazil	*A. sculptum*	0.5% (1/200)	Dragging	[140]
Chile	*Ornithodoros* sp.	100% (4/4)	Off-host (rodent)	[141]
Caldas, Colombia	*A. longirostre*, *A. varium*, *Ixodes* sp.	6.67% (3/45), 66.6% (2/3), 11.1% (1/9)	Off-host (wild Birds)	[142]
Villeta, Colombia	*A. cajennense*	9.1% (1/11)	Off-host (humans)	[143]
Costa Rica	*A. cajennense*	66.7% (10/15)	Off-host (horse)	[144]
Costa Rica	*A. mixtum*, *A. ovale*, *D. nitens*, *R. sanguineus*	NA ^g^	Off-host (horses, cows, humans, dogs) and off-vegetation	[145]
Costa Rica	*R. sanguineus*	16.7% (1/6 pools)	Off-host (dogs)	[146]
Costa Rica	*A. longirostre*, *A. geayi*	7.4% (4/54 pools), 3.7% (2/54 pools)	Off-host (wild birds)	[147]
Cuba	*A. mixtum*	35.7% (5/14)	Off-host (horses)	[148]
Cuba	*A. mixtum*, *Amblyomma* spp.	73% pools (422) ^f^	Off-host (horses and dogs)	[149]
El Salvador	*A. mixtum*, *A. parvum*, *D. nitens, Amblyomma* spp.	77% (10/13), 50% (8/16), 8% (1/13), 11% (1/9)	Off-host (wild and domestic animals)	[150]
Regina, French Guiana	*A. coelebs*	15.4% (2/13)	Off-host (white-lipped peccaries)	[151]
French Guiana	*A. cajennense*, many more	25% ^f^, NA ^g,f^	Dragging, Flagging, Off-host	[152]
Honduras	*A. longirostre*, *A. mixtum*	14% (1/7), 80% (4/5)	Off-host (wild birds, humans)	[153]
Reynosa, Mexico	*R. sanguineus*	1% (3/292)	Off-host (domestic dogs)	[154]
Mexico	*A. mixtum*, *A. oblongoguttatum*, *A. parvum*	33.5% (59/176), 3.17% (2/63), 5.7% (3/53)	Off-vegetation, Off-host (possum, ocelot, white-nosed coati)	[155]
Mexico	*A. maculatum*, *A. mixtum*	9.8% (5/51), 13.2% (17/129)	Off-host (wild pigs)	[156]
Mexico	*A. mixtum*	50% (1/2)	Off-host (iguana)	[157]
Mexico	*A. mixtum*	40% (2/5)	Human	[158]
Nicaragua	*A. ovale*	100% (1/1 pools)	Off-host (dogs)	[146]
Pakistan	*R. microplus*, *H. anatolicum*, *and H. dromedarii*	4%(21/514) ^f^	Off-host (buffalo, cattle, sheep, goat, donkey, camel)	[159]
Panama	*A. mixtum*, *H. juxtakochi*, *Amblyomma* spp.	27.8% (5/18), 4.3% (1/23), 8.2% (12/146 pools)	Dragging	[160]
Panama	*A. mixtum*	94.1% (48/51)	Off-host (tapir, horses)	[42]
Panama	*A. cajennense*, *A. oblongoguttatum*, *A. ovale*, *R. sanguineus*	86.4% (38/44), 50% (2/4), 27.9% (12/43), 3.1% (2/64)	Off-host (dogs, horses, pig)	[161]
Panama	*A. cajennense*, *D. nitens*, *R. sanguineus*	36.7% (11/30), 27.4% (17/62), 12.3% (7/57)	Off-host (horses, dogs)	[162]
Panama	*A. mixtum*, *H. Juxtakochi*, Immature *Amblyomma* spp.	27.8% (5/18), 4.6% (3/65), 11.5% (12/104)	Dragging	[160]
Paraguay	*A. longirostre*	11.8% (2/17)	Off-host (wild birds)	[163]

^a^ An infection prevalence % is calculated by dividing the number of ticks (in each species) reported to test positive for *R. amblyommatis* by the total number of ticks collected of that species and multiplying by 100. Values are listed in the respective order of the species listed in the species column. ^b^ Unable to locate the number of ticks infected in publication. ^c^ Unable to report infection prevalence % as not all positive tick samples were sequenced in this study. ^d^ This value is an average infection prevalence over a two-year period. ^e^ The publication did not specify the collection type. ^f^ The total number of ticks collected for each species not reported in the study. ^g^ Unable to calculate infection prevalence % with the information given in the study.

## Data Availability

Not applicable.

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
