# Peer review of "Rickettsia amblyommatis in Ticks: A Review of Distribution, Pathogenicity, and Diversity"

_microorganisms, 2023, doi:10.3390/microorganisms11020493_

Round 1

Reviewer 1 Report

General comments

The review entitled “Rickettsia amblyommatis in Ticks: A Review of Distribution, Pathogenicity, and Diversity” gives an overview about the state-of-the-art in the field of research according to the mentioned spotted fever group Rickettsia. It resumes many ecological, biological and phylogenetical aspects of published literature. The review is well written and facilely understandable. The different studies are well presented and discussed. However, I have some suggestions and commentaries to improve the style and content of the review.

- I would recommend you to include some kind of “Material and Methods” section in the review where you explain how did you achieve and analyze the literature and sequence data (see lines 79-85 and lines 357-362). To have this information in the middle of the main text seems a little bit confusing.

- Tab 1.: Please indicate the size of the analyzed pools and mention if the infection prevalence is the minimum, medium or maximum prevalence.

- Use the scientific name A. americanum instead of “the lone star tick”. Also reduce repetitions; “Amblyomma americanum, also known as the lone star tick” (lines 88-89; 119)

- Please uniform the spelling of “tick-borne”; e.g. lines 31, 38, 40, etc.

- Please organize the species names in alphabetic order; e.g. lines 35-36; 161-164; 383-384; 404-409; etc.

Specific comments:

- lines 40-41: Be careful with this statement. The climate change may change the distribution of tick species but this does not include that more habitats will be hospitable for ticks.

- line 50: Parola et al. 2013 is dividing the genus Rickettsia in four groups named “spotted fever group (SFG) rickettsiae, typhus group rickettsiae, the Rickettsia bellii group, and the Rickettsia canadensis group” (see Merhej V, Raoult D. 2011. Rickettsial evolution in the light of comparative genomics. Biol. Rev. Camb. Philos. Soc. 86:379–405). Please change this information in your manuscript

- line 50: Further, Igolkina et al. (2023) propose thet Candidatus Rickettsia mendelii is a novel separate basal group of the genus Rickettsia. (Igolkina Y, Nikitin A, Verzhutskaya Y, Gordeyko N, Tikunov A, Epikhina T, Tikunova N, Rar V. Multilocus genetic analysis indicates taxonomic status of "Candidatus Rickettsia mendelii" as a separate basal group. Ticks Tick Borne Dis. 2022 Dec 5;14(2)

- lines 56-57: You talk about 29 valid species. Please give references.

- line 93: Please give the scientific name of the ”chiggers”

- lines 100-101: Please make reference to Table 1

- line 110: Please change “isolated from” to “detected in”·

- lines 103ff, 140ff, 144, 150ff, 188-189: Amblyomma americanum, aggressive, human-biting, etc. Please reduce the repetition of information

- lines 174ff Geographic Range: Maybe it would be interesting to include a map with the reported findings of R. amblyommatis in this section

- Why didn’t you used ompB sequences? Roux and Raoult (2000) mentioned that phylogenetic analysis with ompB sequences show more inter and intraspecific differences between different Rickettsia species than analysis done with gltA or ompA. (V. Roux, D. Raoult, Phylogenetic analysis of members of the genus Rickettsia using the gene encoding the outer-membrane protein rOmpB (ompB), Int. J. Syst. Evol. 50 (2000) 1449-1455)

- lines 364-373: Please check the numbers of used sequences and base pairs in the text and the figure 2 and 3.

- Lines 371-372: The gene for the citrate synthase (gltA) has over 1200 bp. As the gene sequence is already very conserved and the information about small phylogenetic divergences is hard to get from gltA: Why did you used only 278bp for the analyses?

- lines 374-381 and figure 2: It seems rare that the sequence of R. amblyommatis (MG887824) from Brazil (Group 5) is so distant of the other sequence groups of R. amblyommatis. Sebastian. et al. (2020) also implemented this sequence in their ompA analyses and the sequence (named as Haplotype 8) is well include in the branch of R. amblyommatis (Sebastian PS, Tarragona EL, Saracho Bottero MN, Nava S. Phylogenetic divergence between Rickettsia amblyommatis strains from Argentina. Comp Immunol Microbiol Infect Dis. 2020 Apr;69:101418. doi: 10.1016/j.cimid.2020.101418). Please check this.

- figures 2 and 3: Please enhance the quality of the figures. Some sequence names and bootstrap values are moved. What does the “n” in sequence names stand for?

- figure 3: Why is the outgroup Rickettsia rickettsii included in the same clade as some R. amblyommatis (Group 3) sequences? Please check this.

- figures S1 and S2: For a better understanding I would include this figure in the main text as Figure 2A and 2B for example.

- figure S1 and S2: Please provide the information about the different groups as done in figure 2 and 3

- figure S1: At the top; please check the bootstrap values (0?, 1?)

- figure S2: I recommend you to delete the sequence of Ca. R. andeanae and just let R. rickettsii as outgroup

Reviewer 2 Report

The manuscript describes in detail the current knowledge of the distribution, tick vectors and pathogenic properties of Rickettsia amblyommatis. In general, the manuscript is well written and will be of interest to readers of the journal.

The disadvantage of this review article is a very long introduction (Sections 1.1 and 1.2); it should be shortened and be closer to the subject of the manuscript. Since R. amblyommatis was found only in Americas, the authors may limit the Introduction to describing the species of ticks and rickettsiae that inhabit the Americas. At the same time, the manuscript lacks important information about the cultivation of R. amblyommatis, the type strains of R. amblyommatis, complete genome sequencing, and the phylogenetic position of R. amblyommatis. This information should be included. However, the most serious problems of the manuscript are related to the description of phylogenetic analysis. The authors incorrectly performed a phylogenetic analysis of the known R. amblyommatis sequences and did not discuss other publications on this topic. Therefore, this section should be substantially corrected.

Specific comments

1. Lines 44-45 - “the number of species of Rickettsia identified has increased significantly over the past 40 years [14,15]” – please include more recent publications.

2. Line 50 – Rickettsia spp. includes also ancestral groups (R. bellii, R. canadensis and “Candidatus R. mendelii” groups).

3. Lines 104-106 “In addition to being ubiquitous, R. amblyommatis is reported to have interesting other properties such as inhibiting tick growth when coinfected with other species of Rickettsia [24,42]…” please correct the phrase. R. amblyommatis does not inhibit tick growth; at least this information is missing in the cited articles.

4. Section 2.2- please include in the manuscript a map showing the distribution areas of R. amblommatis.

5. Lines 187-191 “There are a number of possible reasons for the wide spread distribution of R. amblyommatis including A. americanum’s aggressive and generalist tendencies, but one of the main underlying causes of its wide geographic range is its ability to be transferred transtadially and transovarially in ticks [176,184–186]”. - This is not an explanation, since all the studied SFGR are transferred transtadially and transovarially in ticks.

6. Line 235 –The references are incorrect. Ref [25] describes not R. amblyommatis but another rickettsial species; Ref [195] is not an article of Burgdorfer.

7. Lines 233-251. Different studies have observed different pathogenicity of R. amblyommatis in guinea pigs. This could be due to the fact that different strains of R. amblyommatis were used in the experiments (discussed in the Introduction section in article of Yen et al, 2021 [Ref. 9 in the Ms]).

8. Line 260. Please indicate that in addition to serological detection, R. amblyommatis DNA was found in a single case in the blood of a dog (Barrett et al., 2014 [Ref. 83 in the Ms])

9. Section 3.1. It should be clearly stated that, despite serological confirmation, there is no direct evidence that R. amblyommatis may cause human infection. Please also discuss whether other Rickettsia species (in addition to R. amblyommatis and R. rickettsii) can cause human infection in areas where R. amblyommatis occurs.

10. Lines 289-290. “While there has not been many cases where R. amblyommatis was  isolated from human patients,…” – as described in this Ms and other articles, R. amblyommatis has not been isolated from human patients and has not even been identified in clinical samples by PCR or sequencing.

11. Lines 308-317. Please shorten this paragraph by omitting repetitions

Lines 332-343. This paragraph should be moved to Section 3.1.

 Section 4. Phylogenetic analysis.

Unfortunately, I have a lot of questions to this section.

12. gltA is a conserved gene for SFGR. It can be used to find different haplotypes and determine the relationship between haplotypes and tick species (as in the case of R. raoultii); however, gltA is not a suitable gene for identifying clusters within the same species of SFGR.

13. Very short sequences were used for phylogenetic analysis. Blastn analysis of the sequences analyzed in this Ms revealed at least 28 gltA sequences of R. amblyommatis with a length  >378 bp  (instead of 278 bp) and more than 40 ompA sequences with a length  >487 bp  (instead of 392 bp). Using longer sequences will make the phylogenetic analysis more reliable.

14. ML is considered to be a more reliable method of phylogenetic analysis compared to NJ. Please include ML trees in the text of manuscript; ML trees should be excluded even from Supplementary. Notably, the differences in clustering between NJ and ML trees are due to the fact that clusters are not well-supported.

15. Clusters were not correctly identified. The ompA NJ tree contains only four clusters (clusters and 2 form a common monophyletic group), while the ompA ML tree contains three clusters; moreover, most of these clusters are not well-supported.

16. ompA cluster 5 contains one sequence (MG887824); however, Blastn analysis showed that MG887824 sequence has 99,8% similarity to JF523328 sequence (100% coverage) and similar identity to the shorter MH445972 and OP375584 sequences (83% coverage).

17. Phylogenetic trees should include not only R. amblyommatis sequences, but also sequences of all (or most) validated species belonging to the main SFGR cluster (e.g. see phylogeny of R. amblyommatis in Sebastian et al., 2020 [ref. 192]). Notably, Sebastian et al., 2020 recommended using the ompA and ompB phylogeny for R. amblyommatis delineating.

18. The biggest error in this section is that in both gltA trees, two identical sequences from the same An13 strain were assigned to different clusters (1 and 2). This means that the phylogenetic trees were reconstructed incorrectly. The authors should find the reason of their error and repeat phylogenetic analysis for both genes. Note that all sequences used to reconstruct a phylogenetic tree must be the same length.

19. R. amblyommatis gltA cluster 3 is closer to other SFGR than it is to cluster 1. The Blastn analysis of strain Conduru sequence from cluster 3 showed that it is identical or closely related to some R. amblyommatis and all R. raoultii sequences. This may mean that some of the R. amblyommatis isolates were species misidentified or that the analyzed gltA fragment is not suitable for R. amblyommatis phylogeny.

20. Lines 380-382 “…there is no clear cluster organization based on tick species or the geographic origin of the sequences, which could be due to the sequence length in this analysis”. Does it mean that the used sequences were short, or that sequences of different lengths were analyzed?

21. Please use larger font for isolates naming in phylogenetic trees

In summary, the section “Phylogenetic Analysis” should describe phylogenetic position of R. amblyommatis based on long concatenated sequences (e.g. see in Karpathy et al, 2016). Specify which SFGR species R. amblyommatis is most closely related to.

I recommend that the authors study the intraspecific diversity of R. amblyommatis based on the scheme described in Sebastian et al., 2020, but with more sequences. Alternatively, authors may exclude their own phylogenetic analysis and report only the results of phylogenetic analyses from other publications.

Round 2

Reviewer 1 Report

Line 84: “Spotted fever group (SFG)” was already mentioned

lines 113, 116: Change Rickettsia amblyommatis for R. amblyommatis

line 166: A. americanum instead of Amblyomma americanum

lines 269 ff: Pathogenicity section. Check the spelling of Guinea pigs

line 334: Apperson et al.

line 388: A. americanum instead of the lone star tick

lines 438-439: How can one group cluster includes 56 sequences when you only used 36 sequences as mentioned in line 433? Please check this. See also N in line 446

line 534: Figure 4 A. Please check why R. rickettsii is not forming a clear outgroup

lines 405: Phylogenetic Analysis. As short gltA sequences are not suitable for the clear differentiation of species-sequence clusters (as you wrote in the text and as seen in the figures), I would recommend you to exclude this analysis from the review and just base it on ompA sequences. You can mention it in the text that the sequences of gltA based on different amplification assays and therefor are not comparable. Also you have to mention the fact that ompB sequences are more suitable for the detection of interspecific differences. Figures 4A and 4B should be deleted.

Author Response

Reviewer 1- 2nd Revisions

Reviewer Comments:  Line 84: “Spotted fever group (SFG)” was already mentioned

Author Response: This has been corrected to only say SFG now.

Reviewer Comments: lines 113, 116: Change Rickettsia amblyommatis for R. amblyommatis

Author Response: Done.

Reviewer Comments: line 166: A. americanum instead of Amblyomma americanum

Author Response: Corrected. 

Reviewer Comments: lines 269 ff: Pathogenicity section. Check the spelling of Guinea pigs

Author Response: This mistake has been corrected.

Reviewer Comments: line 334: Apperson et al.

Author Response: This has been corrected, the period has been added to et al.

Reviewer Comments: line 388: A. americanum instead of the lone star tick

Author Response:  Corrected.

Reviewer Comments: lines 438-439: How can one group cluster includes 56 sequences when you only used 36 sequences as mentioned in line 433? Please check this. See also N in line 446

Author Response: The gltA analysis was deleted which was the analysis that contained 36 sequences. These sections have been changed along with the figure numbers and are more clear now.

Reviewer Comments: line 534: Figure 4 A. Please check why R. rickettsii is not forming a clear outgroup.

Author Response: The Figure 4 A has been deleted as recommended by the reviwer 2.

Reviewer Comments lines 405: Phylogenetic Analysis. As short gltA sequences are not suitable for the clear differentiation of species-sequence clusters (as you wrote in the text and as seen in the figures), I would recommend you to exclude this analysis from the review and just base it on ompA sequences. You can mention it in the text that the sequences of gltA based on different amplification assays and therefor are not comparable. Also you have to mention the fact that ompB sequences are more suitable for the detection of interspecific differences. Figures 4A and 4B should be deleted.

Author Response: The phylogenetic analyses of gltA (4A and B) have been deleted and replaced with an analysis on the gene region ompB. Additionally, the neighbor-joining analyses have been deleted and only the maximum likelihood (ML) analyses were included in the manuscript. Another sequence was also added to the ompA analysis, which resulted in some reorganizing of the tree, showing some cluster organization by tick species and geographic location. Additionally we have added a sentence mentioning the importance of analyzing ompB to see intraspecific differences: “Sequences focusing on these two gene regions were selected as they have been noted to be polymorphic, with ompB being suitable in analyzing intraspecific differences of R. amblyommatis [194].” Thank you for your time and efforts in improving this manuscript.

Reviewer 2 Report

The authors properly responded to some of the comments; however, a number of the most significant comments were not taken into account, mainly concerning the description of phylogenetic analysis. Therefore, I had to repeat some of the recommendations from the previous review.

            I strongly encourage authors to include in the manuscript brief information on cultivation of R. amblyommatis; the type strains of R. amblyommatis; complete genome sequencing, and phylogenetic position of R. amblyommatis among other Rickettsia species.

            The previous Comment 13  “ gltA is a conserved gene for SFGR. It can be used to find different haplotypes and determine the relationship between haplotypes and tick species (as in the case of R. raoultii); however, gltA is not a suitable gene for identifying clusters within the same species of SFGR. Author Response: While this is a good suggestion, we did not have another gene target with sequences of longer lengths.”

            Current Reviewer Response: Indeed, Dolz et al. (2019) recommended the use of concatenated ompA and gltA sequences to study the variability of R. amblommatis. However, the gltA gene itself is too conservative for this. On the contrary, Sebastian et al., 2020 [ref. 192] recommended using the ompA and ompB phylogeny to distinguish between R. amblyommatis isolates. The analysis of BlastN made it possible to find about 35 ompB sequences from the same region with a length of more than 600 bp. Thus, I recommend to use ompB phylogeny instead of gltA phylogeny.

             The previous Comment 15.  “ML is considered to be a more reliable method of phylogenetic analysis compared to NJ. Please include ML trees in the text of manuscript; ML trees should be excluded even from Supplementary. Notably, the differences in clustering between NJ and ML trees are due to the fact that clusters are not well-supported. Author Response: The ML trees have now been removed from the supplementary materials and added to the main text.”

            Current Reviewer Response: Sorry, I made a mistake in my previous comment.  I meant that the NJ trees should be excluded from the manuscript, including the Supplementary. Herein, I strongly recommend that the ompA and gltA NJ trees be excluded from the manuscript. Obviously, the use of NJ and ML trees is redundant. Surprisingly, isolate 11-TC-1-1 belonged to cluster 2 in the ompA NJ tree, but in the ompA ML tree, this isolate was identical to a number of isolates from cluster 1. This means that the difference between NJ clusters 1 and 2 is insignificant.

             The previous Comment 17. “ompA cluster 5 contains one sequence (MG887824); however, Blastn analysis showed that MG887824 sequence has 99,8% similarity to JF523328 sequence (100% coverage) and similar identity to the shorter MH445972 and OP375584 sequences (83% coverage). Author Response: We appreciate the reviewer’s comments and concerns. As the reviewer noted, that is true when we did searches directly on the database; however, phylogenic trees have been construed using only the same lengths.”

            Current Reviewer Response: In the revised version of the manuscript, this cluster still contains one sequence. However, the authors should also analyze the JF523328 sequence from the same cluster (1 substitution, 488 bp from the analyzed region). Most likely, the cluster will also contain the sequences MH445972 (100% identity, 457 bp) and OP375584 (1 substitution, 457 bp)

             To the previous Comment 19.

            Current Reviewer Response: The two An13 sequences differ in the analyzed fragment by only one substitution. So An13 (DQ517290) cannot be assigned to a separate gltA NJ cluster 2. Also, I don't understand why str. Conduru and iso BA030 belong to different clusters in the ML tree and to a common cluster in the NJ tree.

             The previous Comment 22. Please use larger font for isolates naming in phylogenetic trees. Author Response: We attempted to increase the size but because the figure is so large it still appears small.

            Current Reviewer Response: Authors may shorten the names of isolates, such as "Ca. R." instead of "Candidatus Rickettsia" and increase the font size. In its present form, the names of isolates are poorly readable.

 Minor remarks

Line 74 “The primary reservoir and vector of SFG Rickettsia is ticks” - Please use the correct singular and plural

Line 369 Please delete ”when”
